# Contextualized Policy Recovery: Modeling and Interpreting Medical Decisions with Adaptive Imitation Learning

## Abstract

Interpretable policy learning seeks to estimate intelligible decision policies from observed actions; however, existing models fall short by forcing a tradeoff between accuracy and interpretability. This tradeoff limits data-driven interpretations of human decision-making process. e.g. to audit medical decisions for biases and suboptimal practices, we require models of decision processes which provide concise descriptions of complex behaviors. Fundamentally, existing approaches are burdened by this tradeoff because they represent the underlying decision process as a universal policy, when in fact human decisions are dynamic and can change drastically with contextual information. Thus, we propose Contextualized Policy Recovery (CPR), which re-frames the problem of modeling complex decision processes as a multi-task learning problem in which complex decision policies are comprised of context-specific policies. CPR models each context-specific policy as a linear observation-to-action mapping, and generates new decision models *on-demand* as contexts are updated with new observations. CPR is compatible with fully offline and partially observable decision environments, and can be tailored to incorporate any recurrent black-box model or interpretable decision model. We assess CPR through studies on simulated and real data, achieving state-of-the-art performance on the canonical tasks of predicting antibiotic prescription in intensive care units ($+22\%$ AUROC vs. previous SOTA) and predicting MRI prescription for Alzheimer's patients ($+7.7\%$ AUROC vs. previous SOTA). With this improvement in predictive performance, CPR closes the accuracy gap between interpretable and black-box methods for policy learning, allowing high-resolution exploration and analysis of context-specific decision models.

## 1 Introduction

Interpretable policy learning (Hüyük et al., 2022) seeks to recover an underlying decision-making process from a dataset of demonstrated behavior, and represent this process as an interpretable model that can be quantified, audited, and intuitively understood. This approach has gained considerable attention in the medical informatics community as a promising approach for improving standards of care by detecting bias, explaining sub-optimal outcomes (Lengerich et al., 2022), and quantifying regional (McKinlay et al., 2007) and institutional (Westert et al., 2018) differences. Classic machine learning algorithms for policy inference are based on inverse reinforcement learning (Ng & Russell, 2000) or imitation learning (Bain & Sammut, 1999; Piot et al., 2014; Ho & Ermon, 2016), and use black-box architectures such as recurrent neural networks. These approaches have been applied in various medical domains, most prominently oncology prognosis (Beck et al., 2011; Esteva et al., 2017). However, black-box methods have been met with skepticism from the medical community based on a lack of interpretability as well as an inability to identify catastrophic failure modes and generalization issues (Laï et al., 2020; Royal Society (Great Britain) and Royal Society (Great Britain) Staff, 2017).

To achieve this desire for interpretable policies, there has been a recent surge in transparent policy parametrizations for imitation learning. Recent approaches include recurrent decision trees (Pace et al., 2022), visual decision boundaries (Hüyük et al., 2022), high-level programming syntax (Verma et al., 2018) or outcome preferences (Yau et al., 2020). These approaches are generally more interpretable to clinicians, but their interpretability stems from restrictive modeling architec-

**Modeling Medical Decisions with Dynamic Treatment Context**

Figure 1: CPR uses patient-specific treatment contexts to estimate the agent's decision model at each timestep. Decision models are context-specific, and each action probability is an interpretable linear combination of observed features. In this way, CPR achieves exact model-based interpretability without sacrificing representational capacity.

tures, sacrificing performance or imposing obscure constraints that make real-world applications challenging. The primary challenge is that human decisions are informed by a variety of factors including patient background, medical history, lab tests, and more, and true human decision processes are complex. Thus, compressing the decision policy into a single universal observation-to-action mapping necessitates the use of large nonparametric models (e.g. neural nets) that preclude direct interpretability, or produce a model which fails to capture the complexities of human decision-making. Succinctly, models must be both accurate *and* interpretable to effectively support clinical decisions.

In this paper, we propose Contextual Policy Recovery (CPR). Instead of seeking a universal policy that necessitates trading off accuracy against interpretability, CPR embraces the wealth of contextual information that guides human decision-making and reframes the problem of policy learning as multi-task learning of interpretable *context-specific* policies. CPR learns a black-box generator function that encodes contextual information (e.g. the history of observed symptoms and actions which have previously occurred in the decision process) and generates linear observation-to-action mappings. With this combination of black-box and glass-box components, CPR provides interpretable context-specific decision functions at sample-specific resolution. CPR does not sacrifice representational capacity and achieves state-of-the-art performance in policy recovery. Finally, CPR is a modular framework: both the model family of the context encoder and the interpretable observation-to-action mapping model can be chosen according to the given task at hand.

**Contributions:** Our work makes the following contributions to the personalized modeling and medical machine learning communities:

- We propose CPR, a framework for estimating time-varying and context-dependent policies as linear observation-to-action mappings, operating in a fully offline and partially observable environment. CPR enables interpretable and personalized imitation learning, dynamically incorporating new information to model decisions over the entire course of treatment.

- We apply CPR to two canonical medical imitation learning tasks: predicting antibiotic prescription in intensive care units and predicting MRI prescription for Alzheimer's patients. CPR matches the performance of black-box models, generating decision models that recover best-practice treatment policies under a continuum of patient contexts.

- We simulate a heterogeneous Markov decision process and who that CPR empiriclaly converges to the true decision model parameters, improving representation, performance, and interpretability over black-box-only models.

## 2 RELATED WORK

| Policy class
Representative model | Contextualized
CPR | Global Tree
POETREE | Global Aggregate
INTERPOLE |
|---|---|---|---|
| $\mathbb{P}(a_t \mid x_{0:t}, a_{0:t-1})$ | $f_{h(x_{0:t-1}, a_{0:t-1})}(x_t)$ | $f_\theta(x_t, h(x_{0:t-1}))$ | $f_\theta(h(x_{0:t}, a_{0:t-1}))$ |
| |  |  |  |
| Adapts to observed actions | ✓ | ✗ | ✓ |
| $x_t \rightarrow a_t$ glass-box | ✓ | ✓ | ✗ |

Table 1: Comparison of different policy learning algorithms. All listed model classes are designed to provide interpretability beyond what a black-box recurrent model would provide: CPR provides contextualized parametric policies, POETREE (Pace et al., 2022) provides a global tree-based policy, and INTERPOLE (Hüyük et al., 2022) provides interpretations of belief states in $h$ as a summary of patient history.

We seek to learn, at each timestep, an interpretable parametrization of observed behavior to understand how an agent's action was taken in a partially observable, offline environment expanding on the objective of classical imitation learning, that solely seeks to replicate demonstrated behavior. CPR combines the strengths of previous work (Table 1) by keeping the observation-to-action mapping at each timestep interpretable while being able to adapt to the full observed past, aligning learned policies closer with demonstrated behavior. While POETREE (Pace et al., 2022) is able to carry over a hidden state through time, this hidden state acts as an additive bias at each timestep instead of adapting the underlying model parameters $\theta$, which are static throughout time.

**Imitation Learning** The classical approach for learning sequential decision-making processes involves reinforcement learning, optimizing an agent's reward signal $R$ in an online environment. However, in some applications, such as clinical decision-making, experiments with online policies would be both unethical and impractical so only observational data is available. In this setting we do not have access to the reward signal $R$ and instead focus on the inverse task of replicating the observed behavior of an agent, known as *imitation learning*. There are several approaches to tackle such problems, including simple behavioral cloning, without taking interpretability into account, in which the task is reduced to supervised learning mapping observations to actions (Bain & Sammut, 1999; Piot et al., 2014; Sun et al., 2017). Other approaches are based on distribution matching where adversarial training is used to match the state-action distributions between demonstrator and learned policies (Ho & Ermon, 2016; Jeon et al., 2018; Kostrikov et al., 2020). Adaptations of inverse reinforcement learning have been proposed for partial observability (Choi & Kim, 2011) and offline learning (Makino & Takeuchi, 2012). These approaches solely utilize black-box models to maximize action-matching performance, making it challenging to distill a transparent and tractable description of the learned policies.

**Interpretable Policy Learning** Recurrent neural networks depend on latent hidden representations for each feed-forward pass in a time series, obfuscating popular post-hoc explanation methods like LIME (Ribeiro et al., 2016) and SHAP (Lundberg & Lee, 2017). Instead, recent interpretable policy learning methods have focused on adding limiting assumptions, although this also limits real world applicability. INTERPOLE (Hüyük et al., 2022) is a notable example, which parameterizes a latent belief space that relates to decisions, but falls short of explaining how this belief relates to prior observed information. Another notable method is POETREE (Pace et al., 2022), which parameterizes policies as a recurrent soft decision trees (Frosst & Hinton, 2017). While this approach results in a human-interpretable model which maps between timeseries observations and actions, POETREE requires significant post-processing to remove uninterpretable components critical to training. This post-processing step sacrifices performance, especially if the observation space is high-dimensional. Further, these methods are often less accurate at decision modeling than workhorse models like logistic regression (Table 1).

## 3 METHODS

### 3.1 PRELIMINARIES: CONTEXTUALIZED MODELING

Given a dataset consisting of targets $y \in Y$, observations $x \in X$ and context $c \in C$, with the corresponding random variables denoted as $\mathbf{Y}$, $\mathbf{X}$ and $\mathbf{C}$, we want to learn a model $\mathbb{P}(\mathbf{Y}|x, c, \theta)$ predicting $y$ from $x$ and $c$. With this, the probabilistic model is defined as follows:

$$y \sim \mathbb{P}(\mathbf{Y}|x, \theta), \quad \theta \sim \mathbb{P}(\theta|c), \quad \mathbb{P}(\mathbf{Y}|x, c) \sim \int_\theta \mathbb{P}(\mathbf{Y}|x, \theta) \mathbb{P}(\theta|c) d\theta$$

as described by Al-Shedivat et al. (2020). This formulation allows us to model $\mathbb{P}(\theta|c)$ by any black-box model while keeping $\mathbb{P}(\mathbf{Y}|x, \theta)$ in a simple model class parametrized by $\theta$.

Due to an increase in dataset complexity, heterogeneity, and size, sample-specific inference has driven interest in many application areas (Ageenko et al., 2010; Buettner et al., 2015; Fisher et al., 2018; Hart, 2016; Ng et al., 2015). Contextualized modeling has been used in several different frameworks to estimate the context-specific parameters $\theta$ using a meta-model $\mathbb{P}(\theta|C)$ that relates contextual information $C$ to variation in $\theta$. Typically, $\mathbb{P}(\theta|C)$ is a Diroc delta function $\delta(\theta - h(C))$, where $h(C)$ is a deterministic context encoding. The context encoder $h$ is parameterized as $h(C) = Z^T Q$, where $Q$ controls the rank of the context-specific model space and $Z$ is a deterministic encoding $Z = g_\phi(C)$. Many well-known contextualized models fit this framework: in varying coefficient models (Hastie & Tibshirani, 1993), $g$ is a linear model and $Q$ is the identity matrix, in contextualized explanation networks (Al-Shedivat et al., 2020), $g$ is a deep neural network, and $\|Z\| = 1$. In CPR, $g$ is a differentiable, recurrent history encoder, and $Q$ is the identity matrix.

### 3.2 CONTEXTUALIZED POLICY RECOVERY

CPR builds on recent developments (Pace et al., 2022) in interpretable, offline policy learning. Let dataset $D = \{(x_1^i, a_1^i), ..., (x_{T_i}^i, a_{T_i}^i)\}_{i=1}^N$ consist of $N$ treatment trajectories, where each patient $i$ is observed over $T_i$ discrete timesteps for symptoms $x \in \mathcal{X}$ and physician actions $a \in \mathcal{A}$. The data is generated by an unknown policy of the physician agent $\mathbb{P}(a_t|x_1, a_1, ..., x_{t-1}, a_{t-1}, x_t)$ where the action probability at time $t$ is a function of the agent's current state, which is defined by the current and past patient symptoms, as well as past actions.

To recover a policy that is both tractable and interpretable, CPR makes a practical and clinically-minded assumption that a physician places the highest importance on the most current patient information when deciding an action. While treatment history is important, this history is primarily useful for placing the patient's current disease presentation within a context of disease progression and past treatment attempts. To represent this information hierachy, CPR leverages contextual and historical features to generate context-specific policy models.

$$P(a_t|x_1, a_1, ..., x_{t-1}, a_{t-1}, x_t) := f_{\theta_t}(a_t|x_t)$$
$$\theta_t := g(x_1, a_1, ..., x_{t-1}, a_{t-1})$$

Where $f$ is an interpretable model class, e.g. logistic regression, parameterized by a context-specific $\theta$, and $\theta$ is generated via a historical context encoder $g$. The effects of current observation $x_t$ on action probabilities $a_t$ are directly explained through the simple context-specific model $f_\theta$. Furthermore, $g$ can take any functional form without precluding the interpretability of $f$. The context-specific policy models $f_{\theta_t^i}$ are generated for each patient $i$ at each timepoint $t$, allowing us to investigate how previous actions, previous symptoms, patient covariates, and treatment time influence the policy. CPR flexibly allows the context encoder $g$ and the observation-to-action function $f$ to be freely chosen, although they must be differentiable to allow for joint optimization under an appropriate loss $\ell$.

$$\min_g \frac{1}{N} \sum_i \frac{1}{T_i} \sum_t \frac{1}{|\mathcal{A}|} \sum_{\widehat{a} \in \mathcal{A}} \ell(a_t^i, f_{g(x_1^i, a_1^i, ..., x_{t-1}^i, a_{t-1}^i)}(\widehat{a}, x_t^i))$$

In our experiments, $g$ is parametrized by either a vanilla RNN or LSTM (Hochreiter & Schmidhuber, 1997), $f$ is a logistic function, actions $\mathcal{A} := \{0, 1\}$ are binary, and $\ell$ is binary cross-entropy loss. Finally, CPR applies a lasso regularizer to $\theta$ to learn robust policy parameters.

## 4 EXPERIMENTS

We apply CPR to recover time-varying, context-specific decision models within complex decision-making processes. First, we evaluate CPR on real MRI prescription data for dementia patients and

antibiotic prescription data in intensive care units. Follow-up analysis of the contextualized models reveals best-practice treatment plans for both common and outlier patients, while also recovering unexpected and meaningful heterogeneity in physician policies. We further ensure CPR's ability to recover true policy models through a simulated heterogeneous Markov decision process.

## 4.1 MEDICAL DATASETS

We apply CPR to two medical datasets for canonical imitation learning tasks, ADNI and MIMIC-III. These datasets are a prime example of partially-observable decision environments in which we are forced to learn from demonstrated behavior, and where learned policies have the potential to improve clinical operations. Empirical results show that CPR significantly outperforms other interpretable baseline models, and even performing on-par with fully black-box models for both EHR datasets. Low brier scores indicate that CPR is well calibrated while achieving SOTA AUROC and AUPRC.

### 4.1.1 MIMIC ANTIBIOTICS

We look at 4195 patients in the intensive care unit over up to 6 timesteps extracted from the Medical Information Mart for Intensive Care III (Johnson et al., 2016) dataset and predict antibiotic prescription based on 7 observations - temperature, hematocrit, potassium, white blood cell count (WBC), blood pressure, heart rate, and creatinine. We removed the hemoglobin feature used in previous work by Pace et al. (2022) since it is highly correlated (>0.95) with the hematocrit feature.

**Contextualized policies reveal and explain heterogeneity in medical decision processes**    To see how decision functions change under different contexts, we compare them in *model space*. UMAP embeddings of the coefficient vectors (Figure 2b) reveal three distinct clusters of decision functions. The rightmost cluster contains the initial model parameters $\theta_0$ for each trajectory. Since there is no context that could differentiate the agent's behavior at the initial visit, these parameters are the same for all patients, and the contextualized models recover the population estimator. Subsequent to this initial homogeneity, heterogeneity in decision policies arises. The larger driver of this heterogeneity is prior antibiotic prescription – patients that previously got antibiotics are more likely to continue to receive antibiotics, while patients that did not receive antibiotics are likely to continue to not receive antibiotics. The lower cluster contains mostly (99.8%) models in which the patient did get antibiotics in the previous state $t-1$, while the upper cluster contains patients (99.3%) that did not get antibiotics in $t-1$. This strong split is only recovered by contextualized policies; global policies that ignore context fail to identify this heterogeneity (Figure 2a). We train two models conditioned on their respective contexts, removing even more variability by limiting observations to the second day in the ICU. The global model only represents a small part of the population. Conditioning on the main driver of model heterogeneity (whether or not a patient got antibiotics in the previous visit) and training individual models for each case yields models that look like an average over the contextualized models of the respective clusters.

To uncover typical treatment regimes, we cluster patients into 5 subgroups over the first 4 days of the ICU stay using hierarchical clustering (Fig. 12). To identify the drivers of this heterogeneity

| Algorithm | ADNI MRI scans | | | MIMIC Antibiotics | | |
|---|---|---|---|---|---|---|
| | AUROC | AUPRC | Brier ↓ | AUROC | AUPRC | Brier ↓ |
| □ Logistic regression | 0.66 ± 0.01 | 0.86 ± 0.00 | 0.16 ± 0.00 | 0.57 ± 0.01 | 0.80 ± 0.01 | 0.20 ± 0.00 |
| □ INTERPOLE † | 0.60 ± 0.04 | 0.81 ± 0.08 | 0.17 ± 0.05 | NR | NR | NR |
| □ INTERPOLE ‡ | 0.44 ± 0.04 | 0.75 ± 0.09 | 0.19 ± 0.07 | 0.65 ± (≤ 0.04) | NR | 0.21 ± (≤ 0.04) |
| □ POETREE ‡ | 0.62 ± 0.01 | 0.82 ± 0.01 | 0.18 ± 0.01 | 0.68 ± (≤ 0.04) | NR | 0.19 ± (≤ 0.04) |
| □ CPR-RNN (ours) | **0.72 ± 0.01** | **0.88 ± 0.01** | **0.15 ± 0.00** | **0.82 ± 0.00** | **0.90 ± 0.00** | **0.14 ± 0.00** |
| □ CPR-LSTM (ours) | **0.72 ± 0.01** | **0.88 ± 0.01** | **0.15 ± 0.00** | **0.82 ± 0.00** | **0.90 ± 0.00** | **0.14 ± 0.00** |
| ■ RNN | 0.72 ± 0.01 | 0.88 ± 0.01 | 0.15 ± 0.00 | 0.83 ± 0.00 | 0.90 ± 0.00 | 0.13 ± 0.00 |
| ■ LSTM | 0.71 ± 0.01 | 0.88 ± 0.01 | 0.15 ± 0.00 | 0.84 ± 0.00 | 0.91 ± 0.00 | 0.13 ± 0.00 |

Table 2: Action-matching performance of imitation learning algorithms. Bolded values denote the best performance of interpretable models. Open source task-agnostic models are reported as mean ± standard error for 10 bootstrap runs. INTERPOLE and POETREE, which are task-specific or closed-source, are based on prior reports: † reported by Hüyük et al. (2022), ‡ reported by Pace et al. (2022). NR: No values reported. □: interpretable method. ■: black-box method.

in decision policies, we examine the parameters of the decision function as a function of context (Figure 2c). The most notable difference in parameters is the intercept value which is positive for the group of patients that get treated with antibiotics and negative for patients that did not receive antibiotic treatment. This is consistent across timesteps, and it is likely that a patient is prescribed antibiotics if they got it the day before. This aligns with medical protocols that rarely suggest antibiotic treatments briefer 5 days (Guleria et al., 2019).

Previous work by Pace et al. (2022) and Bica et al. (2021) described a patient's temperature and white blood cell count (WBC) as the main drivers of antibiotic treatment decisions since these are known medical criteria to counter infections (Masterton et al., 2008). Our results paint a more nuanced picture, in which heterogeneous coefficients reflect heterogeneous priorities when designing treatment plans. CPR identifies that the influence of temperature changes based on prior antibiotic prescription. For patients who have already been prescribed antibiotics, an infection has already been detected and the doctor's decision model shifts towards mitigating the risk of possible side effects of the antibiotics treatment rather than strictly considering the benefits of the new treatment. This shift in priorities is supported by the change in the creatinine coefficient (Figure 2c). High serum creatinine can be an indicator of impaired kidney function (Gounden et al., 2023), a possible adverse effect of antibiotics

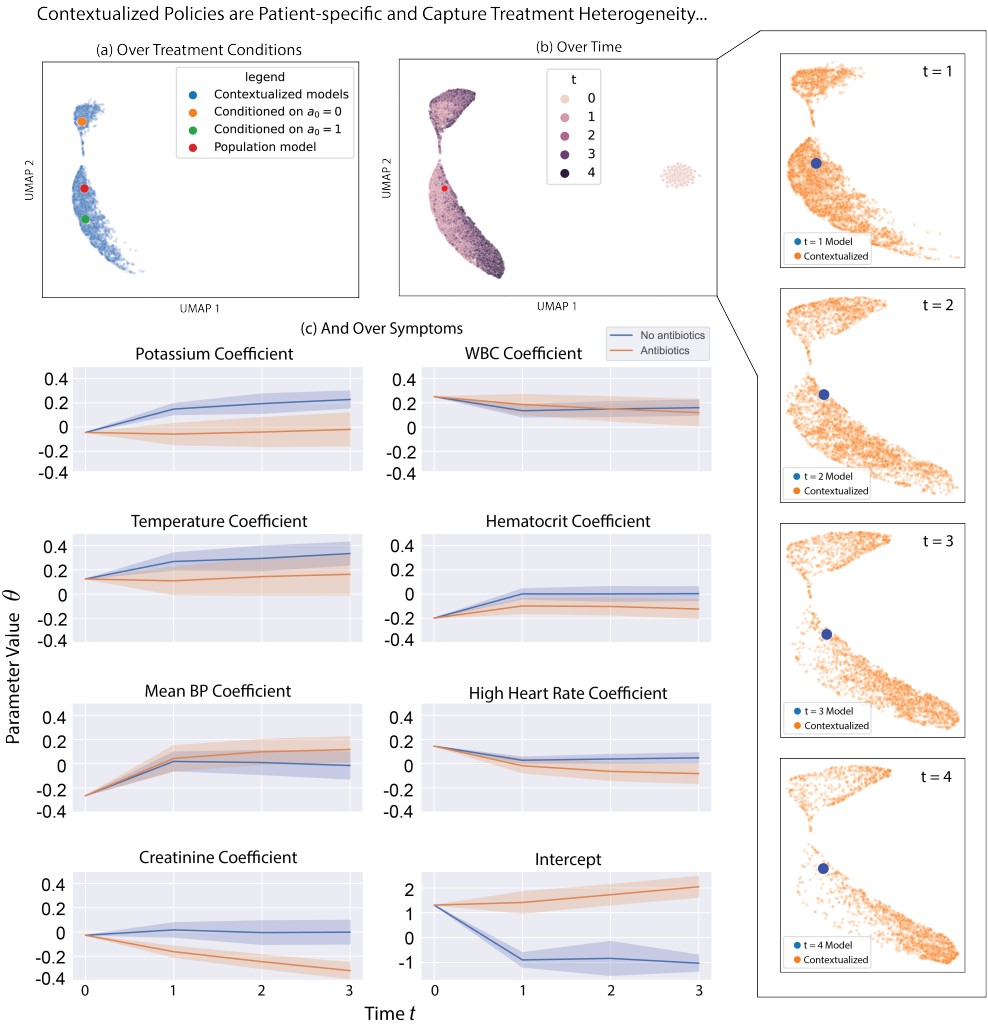

Figure 2: Exploration of contextualized policies generated by CPR for predicting antibiotic prescription. (a) Contextualized policies identify prior antibiotic prescription and (b) treatment time as drivers of treatment heterogeneity. (c) CPR generates policies that evolve with time and treatment history, revealing the context-specific importance of patient symptoms toward future treatments.

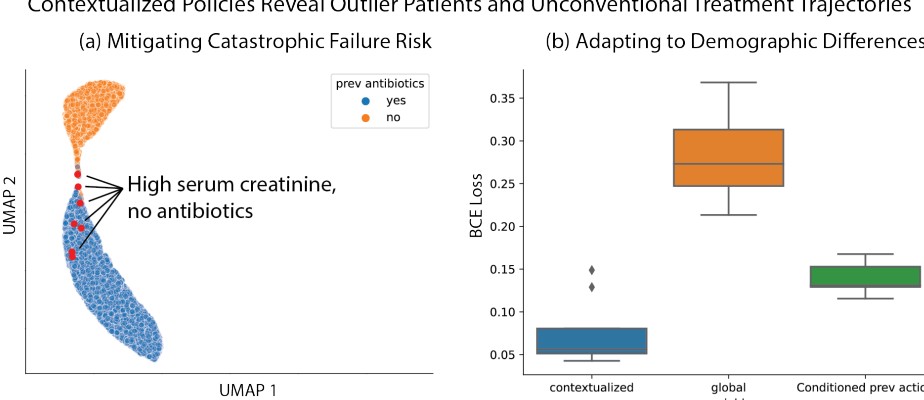

Figure 3: CPR generates decision models for marginal groups with high accuracy. Left: Using only a small subgroup of patients making up 7 observations in the training set, CPR identifies elevated creatinine as a severe risk factor for kidney failure and reassigns patients to a non-antibiotics treatment plan, while these patients would otherwise be likely to receive treatment. Right: For the small subgroup of patients below 20 years of age (with only 9 observations in the held-out set and 44/12 in the train/validation set), CPR improves drastically in terms of cross-entropy loss.

(Khalili et al., 2013); as such, a high creatinine level decreases the probability of continuing antibiotic treatment. Finally, we see that increased potassium is associated with the decision to begin antibiotics (Figure 2c). Electrolyte balance (in which potassium plays a vital role) is an important factor in infectious diseases, and reduced sodium (a 1+ ion competitor with potassium) in particular is known as a marker of viral and bacterial infections (Królicka et al., 2020).

**Contextualized Policies Reveal Outlier Patients**  By relating modeling tasks through task-specific contexts, CPR learns to generate context-specific models even when the number of samples per context is as small as one, and similarly generalizes to generate models for unseen contexts. Previously, we assessed these models in aggregate to reveal common treatment trajectories and best practices (Figures 2b, 9). Here, we demonstrate how personalized models also reveal small subgroups of patients or even individuals with outlier policies. In particular, some patient populations have higher dose tolerances and are amenable to more aggressive treatment, while others display rare comorbidities and risk factors prohibit common treatment options. Personalized treatment options are critical, especially when more common treatment plans need to be avoided.

CPR identifies several of these outlier patients when in the MIMIC antibiotic prescription dataset (Fig. 3). First, younger patients often have fewer comorbidities and more robust immune systems, and physicians can be more confident that antibiotics will not impose any adverse side effects if an infection is suspected. We observe that contextualized policies recover this case, and represent treatment for the under 20 age group much more accurately when antibiotics are prescribed. Second, elevated creatinine is a rare side-effect of antibiotics but a likely indicator of suboptimal Kidney function and possible Kidney failure (Gounden et al., 2023). CPR identifies that patients with elevated creatinine are immediately removed from antibiotics following an initial prescription, placing them in a treatment cluster characterized by a lack of antibiotics prescription that would otherwise be unlikely for these patients. The patient-specific policies produced by CPR provide a novel view of the treatment process, allowing us to easily identify these rare and outlier effects in terms of context-specific policy parameters and errors, revealing nuances in treatment decisions that were missed by prior works.

### 4.1.2 ADNI MRI Scans

Following prior works by Hüyük et al. (2022) and Pace et al. (2022), we apply CPR to 1605 patients from the Alzheimer's Disease Neuroimaging Initiative (ADNI). The canonical task is to predict at each visit whether a Magnetic Resonance Image (MRI) scan is ordered for cognitive disorder diagnosis (noa, 2018). Patient observations consist of the Clinical Dementia Rating (CDR-SB) on a severity scale (normal; questionable impairment; severe dementia) (O'Bryant et al., 2008) and the

Contextualized Policies Capture Policy Heterogeneity Over Static Contexts

Figure 4: Comparison of patient-specific model parameter distributions by age and gender in visit $t = 0$ after incorporating static contexts. Static contexts help to personalize initial models when no history is available.

MRI outcome of the previous visit falling into 4 categories: No MRI scan, below average, average and above average hippocampal volume.

The small number of discrete context features and the checklist-like time-independent diagnostic criteria for Alzheimer's (O'Bryant et al., 2008) provide a limited view of Alzheimer's diagnosis that is unlikely to explain any heterogeneity in clinical decisions. This is reinforced by the fact that a single logistic regression outperforms all interpretable policy baselines (Table 2). We introduce a new condition-specific logistic regression baseline, where we learn a decision model for every set of unique context features at each timestep. Indeed, this condition-specific model performs nearly as well as CPR and black-box models, with an AUROC of 0.71. While CPR and the black-box baselines can still capture dependencies on past actions and observations (Fig. 8, 9), this seems to confer only marginal modeling improvements for the canonical ADNI task.

Instead, we reformulate this canonical task to include a new source of heterogeneity with clinical significance: patient age and gender (Castro-Aldrete et al.). Condition-specific logistic regression and both interpretable baselines (Hüyük et al., 2022; Pace et al., 2022) are unable to model changes in patient-specific policies over continuous static contexts like age, but CPR is able to easily incorporate static as well as dynamic contexts by encoding static contexts into the initial hidden state of the context encoder $g$.

Figure 4 shows how the estimated policies at $t = 0$ differ between four patient subgroups. We find meaningful heterogeneity in the models generated by CPR, where age dominates CDRSB coefficients and overall intercept, while gender dominates hippocampal volume coefficients. Additionally, static contexts substantially increase the action-matching performance of CPR to $0.763$ AUROC.

## 4.2 SIMULATIONS

CPR incorporates both a deep learning component and a statistical modeling component to introduce a novel mechanism of interpretability, the context-specific linear policy. Our approach differs substantially from prior interpretable methods by including a deep learning component. Naturally, we wonder if CPR's explicit linear policy representation is key to its performance and interpretability, or if accurate and robust context-specific linear policies can also be recovered from black-box policy models using post-hoc interpretation methods. To test this, we simulate a heterogeneous, action-dependent Markov decision process (MDP) and evaluate CPR versus black-box baselines on their recovery of true simulation parameters: the true action probability and the true coefficients of a context-specific linear policy (Fig. 5). While CPR explicitly generates these context-specific linear coefficients, black-box models implicitly model these coefficients as feature gradients (i.e. linear coefficients in a first-order Taylor expansion). Akin to popular post-hoc interpretability methods like

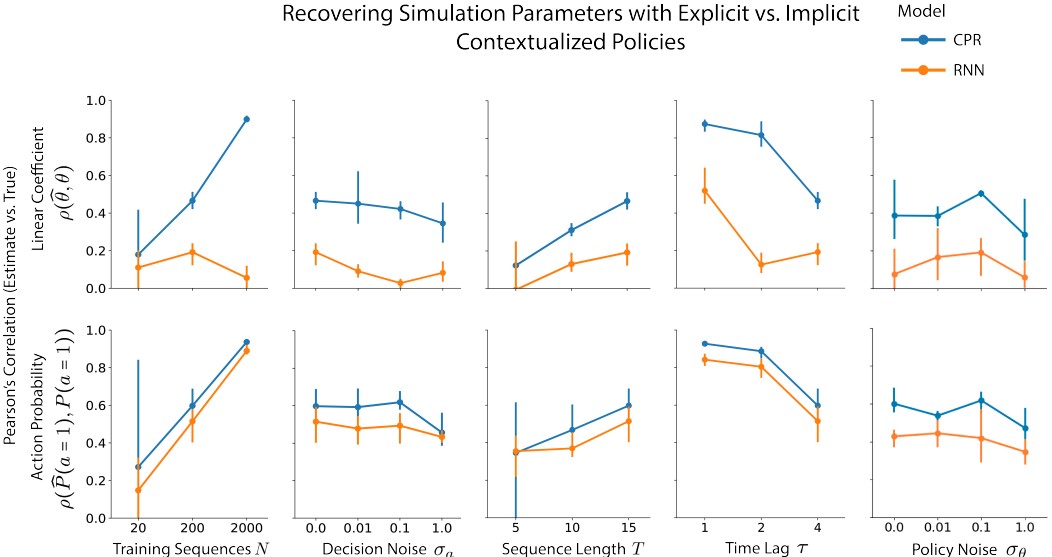

Figure 5: Comparing policy models learned by CPR and RNN in terms of the Pearson's correlation between estimated and true action probabilities and context-specific policy coefficients. We choose default simulation arguments $N = 200$, $\sigma_a = 0$, $T = 15$, $\tau = 4$, and $\sigma_\theta = 0$, varying each parameter individually. We hold out 15% of trajectories at random for evaluation. Results are the mean and 95% confidence interval from three randomly initialized and independently simulated data sets.

LIME (Ribeiro et al., 2016), we leverage the differentiability of RNNs $\Phi(x_t, h) \to a_t$ to recover the implicit context-specific linear policies $\theta$.

$$\widehat{\theta} = \frac{\partial}{\partial x_t}\Phi(x_t, h)$$

We generate data with a known heterogeneous MDP governed by a true context-specific policy

$$P(a_t = 1|x_t, x_{t-\tau}, a_{t-\tau}, t) = 1/(1 + \exp(-\theta \cdot x_t + \epsilon_a)), \qquad \theta = x_{t-\tau} \cdot (2a_{t-\tau} - 1) + \frac{t}{T} + \epsilon_\theta$$

where $T$ is the sequence length or maximum timestep, and $\tau$ is a time lag between the current policy and a dependence on past observations $x_{t-\tau}$ and actions $a_{t-\tau}$. We simulate $N$ total sequences, drawing observations $x_t \sim \text{Unif}[-2, 2]$, policy noise $\epsilon_\theta \sim N(0, \sigma_\theta^2)$, and decision noise $\epsilon_a \sim N(0, \sigma_a^2)$ at each timestep. On a known heterogeneous and action-dependent MDP, CPR's explicit policy representation not only improves its representation of MDP parameters but increases overall performance versus a black-box model with an unstructured policy representation (Fig. 5).

## 5 DISCUSSION

In this study, we propose contextualized policies as dynamic, interpretable, and personalized linear decision models, each representing a single step in a complex treatment process. By relating individual modeling tasks through patient-specific histories and contexts we avoid the pitfalls of common personalization methods that reduce statistical power (e.g. sample splitting and subpopulation grouping). As a result, CPR matches the performance of black-box models while retaining the interpretability of linear models. Post-hoc analysis of patient-specifc models generated by CPR reveal rare covariates with outsize effects on treatment decisions, as well as extremely subtle effects in the general population. While we apply CPR in offline and partially observable environments, CPR is directly portable to online policy inference with only subtle training modifications. As a caveat, CPR makes an explicit trade-off to enable context-specific interpretability that precludes interpreting direct effects of historical features on current actions, which may make it unsuitable for tasks where the effect of historical features are direct and not mediated by a context-specific model. However, for medical imitation learning, CPR's policy representation is well-aligned with clinical practice, where patient history contextualizes current patient symptoms and treatment options. CPR is a step toward reinforcement learning agents that explain as they think, and promises a general purpose platform for supporting and improving complex human decisions across disparate domains.

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

## A  APPENDIX

Code is available on `https://github.com/contextpr/contextualized_policy_replication`. All model embeddings are done with UMAP (McInnes et al.).

### A.1  DATA

-5pt Each dataset is split up into a training set (70% of patients), validation set (15% of patients) for hyperparameter tuning, and test set (15% of patients) to report model performance.

#### A.1.1  ADNI

We follow the task described by Pace et al. (2022) and Hüyük et al. (2022), taking the preprocessing code from Hüyük et al. (2022). Overall, the dataset contains patients with at least two and a median of three visits. We exclude patients without CDRSB measurements in one of their visits.

#### A.1.2  MIMIC

We follow the task of predicting antibiotic prescription in the intensive care unit over up to 6 timesteps as described by Pace et al. (2022) adapting the preprocessing code provided by Jarrett et al. (2021). Patient trajectories with missing values in one of the observed features or non-consecutive measurements, patients above 100 years or below 1 year of age, and short stays below 3 days were eliminated. We end up with 4195 patients in our dataset. Input features were standardized and the hemoglobin feature used in previous work was removed since it is highly correlated with the hematocrit feature ($> 0.95$). See Table 3 for the results using all 8 input features. Since measurements are taken as averages over each day in the ICU, we use the measurements of the previous day as observations $x_t$ to predict $a_t$ ensuring that we don't predict actions from measurements taken after the patient was treated with antibiotics earlier in the day.

| Class | Algorithm | MIMIC Antibiotics | | |
|---|---|---|---|---|
| | | AUROC | AUPRC | Brier $\downarrow$ |
| Interpretable | Logistic regression | 0.60 | 0.79 | 0.20 |
| | INTERPOLE Hüyük et al. (2022) † | NR | NR | NR |
| | INTERPOLE Hüyük et al. (2022) ‡ | 0.65 | NR | 0.21 |
| | POETREE Pace et al. (2022) ‡ | 0.68 | NR | 0.19 |
| | CPR-RNN (ours) | 0.79 | 0.88 | 0.15 |
| | CPR-LSTM (ours) | 0.80 | 0.88 | 0.15 |
| Black-Box | RNN | 0.81 | 0.89 | 0.14 |
| | LSTM | 0.82 | 0.89 | 0.14 |

Table 3: Action-matching performance of different policy learning algorithms on the MIMIC antibiotics task using all 8 input features. Listed performance of INTERPOLE and POETREE are from prior reports of state-of-the-art performance on these canonical datasets: † reported in Hüyük et al. (2022), ‡ reported in Pace et al. (2022). NR: No values reported.

### A.2  IMPLEMENTATION

CPR implements $g$ as a recurrent neural network (either vanilla RNN or LSTM) with a hidden state of dimension $k$ and $h$ as a neural network with one hidden layer, again of size $k$, mapping to the output parameters $\theta$. The initial context $c_0$ at timestep $t = 0$ is set to all zeros.

Black-Box RNN and LSTM are implemented similarly, with $k$ being the dimensionality of the hidden state and one hidden layer of size $k$ directly mapping to the predicted actions $a_t$. The black-box model predicts $P(a_t|x_t, h_t)$ by taking $[x_t, a_{t-1}]$ as input at each timestep and is optimized using the binary cross entropy loss.

## A.3 TRAINING

We train all models using the Adam optimizer Kingma & Ba (2014) and early stopping on the validation set. The initial learning rate chosen for CPR is 5e-4 and 1e-4 for the baseline RNNs. We select the dimensions of the hidden state for both CPR and the baseline RNNs from $[16, 32, 64]$. For CPR, $\lambda$ is chosen from $[0.0001, 0.001, 0.01, 0.1]$. The batch size is selected as $64$ for all models. Table 4 shows the optimal hyperparameters chosen based on the validation set performance.

| | | ADNI MRI scans | | MIMIC Antibiotics | |
|---|---|---|---|---|---|
| Class | Algorithm | $\lambda$ | $k$ | $\lambda$ | $k$ |
| Interpretable | CPR-RNN (ours) | 0.0001 | 32 | 0.0001 | 32 |
| | CPR-LSTM (ours) | 0.001 | 64 | 0.0001 | 32 |
| Black-Box | RNN | - | 64 | - | 32 |
| | LSTM | - | 64 | - | 64 |

Table 4: Hyperparameters chosen for different models.

## A.4 ADDITIONAL EXPERIMENTS

### A.4.1 SIMULATIONS

Figure 6 shows that CPR is able to recover context-dependent threshold decision boundaries where an agent takes an action if the observed value is above or below a certain threshold. Here $x_t$ is sampled as $x_t \sim \text{Unif}[0, 1]$.

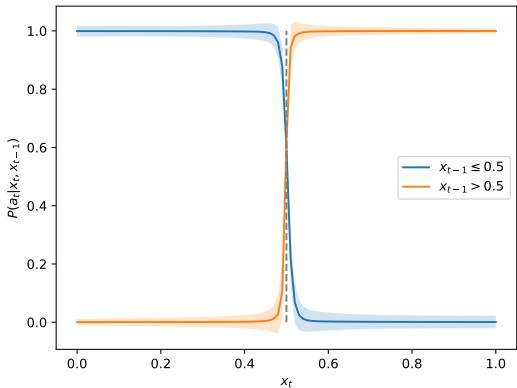

Figure 6: CPR recovers hard decision boundaries, generated by rule-based decision making over time. Here, when the previous observation $x_{t-1} < 0.5$, the current action is taken if $x_t < 0.5$. If $x_{t-1} \geq 0.5$, the current action is reversed. The probabilistic models of these boundaries align closely with the true step function.

We further simulate dynamic, history-dependent decision policies in a homogeneous MDP and evaluate CPR based it's recovery of true simulated true parameters in these policies. We generate $n = 2000$ patient trajectories of length $T = 9$ time steps. At each time step, we sample a random observation variable $x_t \sim \text{Unif}[-1, 1]$ and an agent which takes an action $a_t \in [0, 1]$. This action is determined by a true observation-to-action mapping function which depends on the observed history and total treatment time.

$$P(a_t | x_t, x_{t-1}) = \sigma(w^t(x_{t-1}) * x_t + b^t(t)), \tag{1}$$

where $w^t(x_{t-1}) = 4x_{t-1}$ and $b^t(t) = \frac{t-5}{4}$. We simulate a true contextual policy and assess recovery of simulation parameters with CPR and a RNN. The probability of taking an action at each time $t$ depends on both the absolute point in the time series, as well as the history of observations

immediately preceding any time point. We design this simulation as a succinct way to demonstrate how simple decision models which depend on both absolute and relative effects over a time series often combine to create a universal policy that is exceptionally complex and difficult to capture in a single model that is invariant to time or context. We evaluate CPR based on it's ability to imitate the observed time-dependent and context-dependent actions and recover the simulated true parameters.

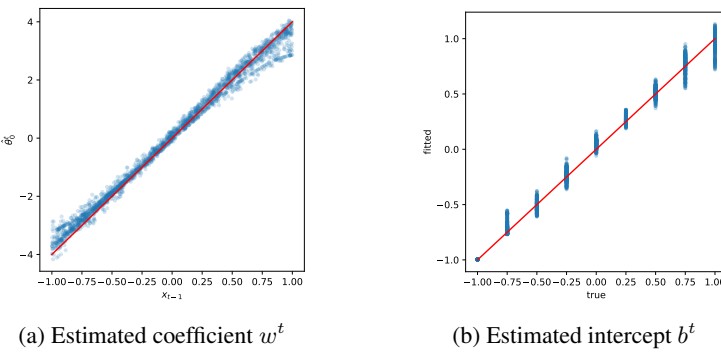

(a) Estimated coefficient $w^t$           (b) Estimated intercept $b^t$

Figure 7: CPR recovers true policy coefficients in a homogeneous MDP

Our method is able to recover the true model parameters (marked in red) as shown in Figure 7 with a slight bias in the upper and lower value range.

### A.4.2   ADNI MRI SCANS

**Bootstrapped Results** We run 10 bootstrap runs to get confidence estimates for model performance on the ADNI (Table 5) dataset. Each bootstrap sample is randomly split into a training, validation and test set.

| | | ADNI MRI scans | | |
|---|---|---|---|---|
| Class | Algorithm | AUROC | AUPRC | Brier ↓ |
| Interpretable | Logistic regression | $0.66 \pm 0.03$ | $0.86 \pm 0.01$ | $0.16 \pm 0.01$ |
| | INTERPOLE † | $0.60 \pm 0.04$ | $0.81 \pm 0.08$ | $0.17 \pm 0.05$ |
| | INTERPOLE ‡ | $0.44 \pm 0.04$ | $0.75 \pm 0.09$ | $0.19 \pm 0.07$ |
| | POETREE ‡ | $0.62 \pm 0.01$ | $0.82 \pm 0.01$ | $0.18 \pm 0.01$ |
| | CPR-RNN (ours) | $\mathbf{0.72 \pm 0.02}$ | $\mathbf{0.88 \pm 0.02}$ | $\mathbf{0.15 \pm 0.01}$ |
| | CPR-LSTM (ours) | $\mathbf{0.72 \pm 0.02}$ | $\mathbf{0.88 \pm 0.02}$ | $\mathbf{0.15 \pm 0.01}$ |
| Black-Box | RNN | $0.72 \pm 0.02$ | $0.88 \pm 0.02$ | $0.15 \pm 0.01$ |
| | LSTM | $0.71 \pm 0.02$ | $0.88 \pm 0.02$ | $0.15 \pm 0.01$ |

Table 5: Action-matching performance of different policy learning algorithms on the ADNI MRI scans task. Bolded values in each column denote the best performance of any interpretable model. Listed performance of INTERPOLE and POETREE are from prior reports of state-of-the-art performance on these canonical datasets: † reported by Hüyük et al. (2022), ‡ reported by Pace et al. (2022).

Figure 8 shows how different contexts influence the agents decision function over time. Most notably, a high CDRSB value during the first two visits decreases the probability of ordering an MRI since this can be already seen as a strong indicator of dementia, making a scan less informative noa (2018). Patients A and B share the same decision model in $t = 1$ since they both show medium CDRSB and avg hippocampal volume in $t = 0$. Afterward, their decision function differs in $t = 2$. Patient A's hippocampal volume was measured as "low" in $t = 1$ leading to a lower overall probability of ordering an MRI in $t = 2$ indicated by a lower intercept and CDRSB coefficients. In contrast, Patient B was again diagnosed with an "avg" hippocampal volume in visit $t = 1$. A low hippocampal volume can again be seen as a strong indicator of dementia making a scan less informative. Patient C, in contrast, is diagnosed with medium CDRSB and high hippocampal volume throughout all visits. This

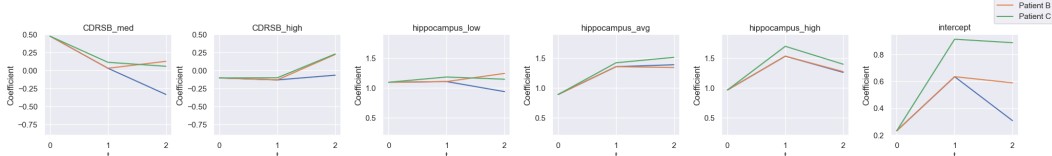

Figure 8: Estimated decision model coefficients over 3 timesteps for 3 patients with representative ADNI contexts. No confidence intervals are available, as the discrete diagnostic features in ADNI produce only a discrete number of possible trajectories.

increases the probability of ordering an MRI (higher intercept) since there is no clear indication that would make an MRI obsolete.

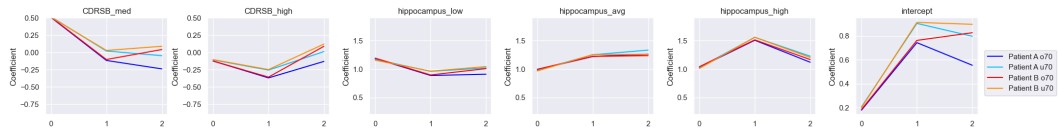

Figure 9: Average estimated coefficients for different ADNI contexts, patients over 70 vs patients under 70. Patient A and B from Figure 8 above. Older patients are less likely to receive MRIs for both patient groups. Static context is age and gender.

**Static Contexts unrolled trough time** Static Contexts such as age and gender do not only influence the observation-to-action mapping at timestep $t = 0$ but uncover heterogeneity across subgroups throughout time. Figure 9 shows that the difference in the intercept coefficient between patients below 70 and above 70 years of age widens over time for both patients A and B indicating that ordering an MRI is less likely for older patients. We can also see a slight difference in the CDRSB_med coefficient with it being slightly negative for old patients, reducing the probability of getting ordering an MRI if an old patient in this patient group scored a medium result, compared to young patients.

### A.4.3 MIMIC ANTIBIOTICS

**Bootstrapped Results** We run 10 bootstrap runs to get confidence estimates for model performance on the MIMIC (Table 6) dataset. Each bootstrap sample is randomly split into a training, validation and test set.

| | | MIMIC Antibiotics | | |
|---|---|---|---|---|
| Class | Algorithm | AUROC | AUPRC | Brier ↓ |
| Interpretable | Logistic regression | $0.57 \pm 0.03$ | $0.80 \pm 0.03$ | $0.20 \pm 0.01$ |
| | INTERPOLE † | NR | NR | NR |
| | INTERPOLE ‡ | 0.65 | NR | 0.21 |
| | POETREE ‡ | 0.68 | NR | 0.19 |
| | CPR-RNN (ours) | $\mathbf{0.82 \pm 0.01}$ | $\mathbf{0.90 \pm 0.01}$ | $0.14 \pm 0.01$ |
| | CPR-LSTM (ours) | $\mathbf{0.82 \pm 0.01}$ | $\mathbf{0.90 \pm 0.01}$ | $\mathbf{0.14 \pm 0.00}$ |
| Black-Box | RNN | $0.83 \pm 0.01$ | $0.90 \pm 0.01$ | $0.13 \pm 0.01$ |
| | LSTM | $0.84 \pm 0.01$ | $0.91 \pm 0.01$ | $0.13 \pm 0.00$ |

Table 6: Action-matching performance of different policy learning algorithms on the MIMIC antibiotics task. Bolded values in each column denote the best performance of any interpretable model. Listed performance of INTERPOLE and POETREE are from prior reports of state-of-the-art performance on these canonical datasets: † reported by Hüyük et al. (2022), ‡ reported by Pace et al. (2022). NR: No values reported.

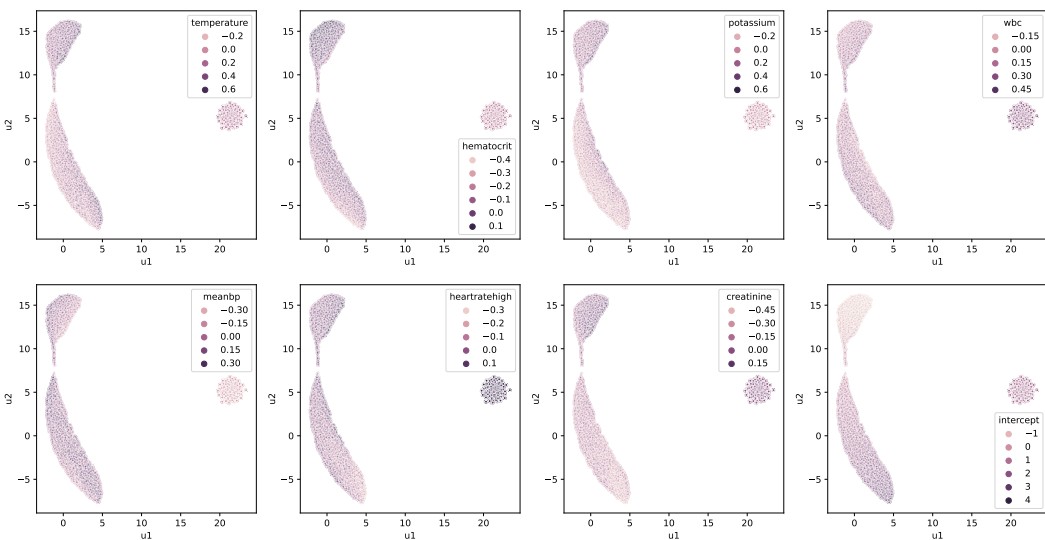

Figure 10: Model coefficient embeddings for MIMIC policy models.

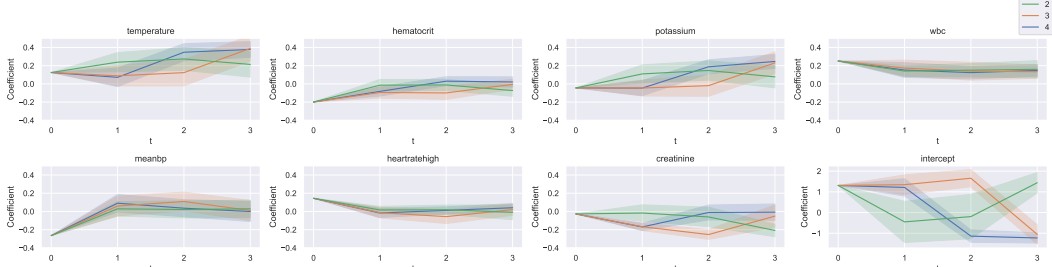

Figure 11: Estimated model parameters for three remaining model trajectories (first two in Fig. 2c). Error bars are coefficient standard deviations across patients in each trajectory group and time point.

**Model Coefficients** Figure 10 shows the heterogeneity in estimated model coefficients. The main drivers of heterogeneity are the intercept, potassium and creatinine coefficients.

**Treatment Trajectories** To uncover typical treatment trajectories, we look at clusters of coefficients over 4 timesteps. We cluster patients into 5 subgroups using hierarchical clustering based on silhouette score ($0.566$) as seen in Figure 12. Figure **??** shows the two largest clusters containing patients that get antibiotics throughout their stay (cluster $1$, 1707 patients) and patients that never get antibiotics (cluster $5$, 384 patients). The remaining three groups are plotted in Figure 11. Patients that get treatment for the first two days fall into cluster $3$ (232 patients). We can see that their treatment parametrization changes significantly after treatment is stopped. Cluster $4$ contains patients that get treated for the first day only (240 patients). Both patients in cluster $3$ and $4$ share similar decision models in $t = 1$ since both were treated in $t = 0$, it diverges in $t = 2$ after treatment is stopped for one group and share the decision parametrization in $t = 3$ after both groups were not treated in $t = 2$. The remaining patients fall into cluster $4$ (375 patients).

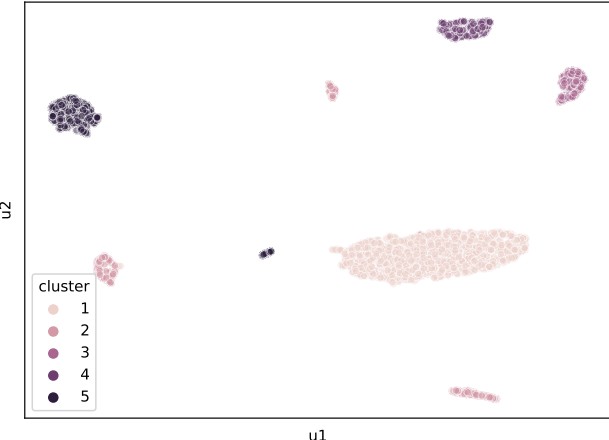

Figure 12: MIMIC patient trajectory clusters, produced by concatenating decision models over 4 consecutive time points into a trajectory matrix and embedding them with UMAP.

# B ADDITIONAL PLOTS

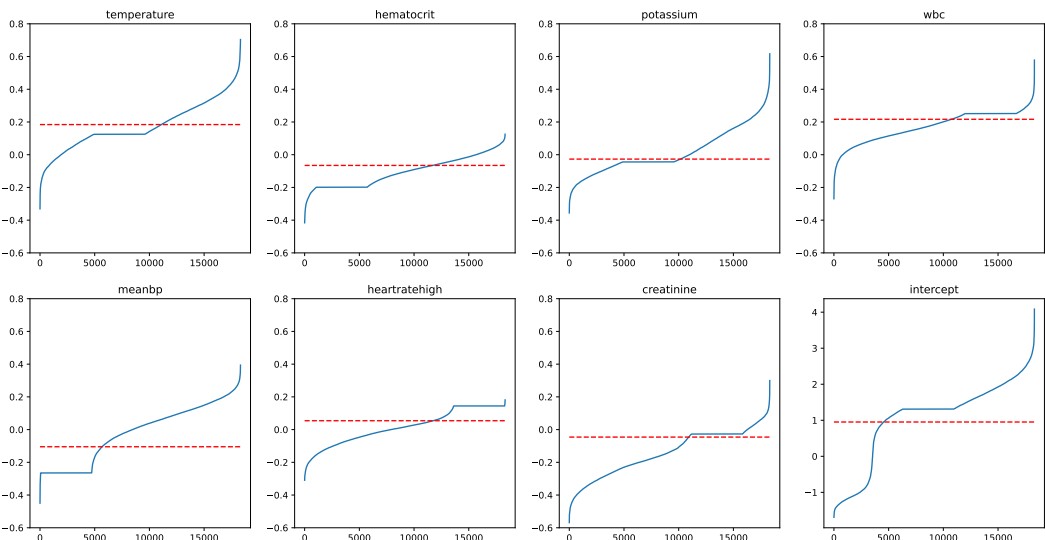

Figure 13: Coefficient values of antibiotic prescription models (MIMIC) parametrized by context vs parameters of global model (dashed red line)

