# OpenReview forum: "Contextualized Policy Recovery: Modeling and Interpreting Medical Decisions with Adaptive Imitation Learning"
_ICLR.cc/2024/Conference — Submitted to ICLR 2024_

### Official Review · Reviewer_TJfx · 2023-10-30

**Soundness:** 3 good
**Presentation:** 4 excellent
**Contribution:** 1 poor
**Rating:** 5
**Confidence:** 4

**Summary:**

The paper proposed an interpretable behavior cloning method for a partially observed MDP, the Contextualized Policy Recovery. It reframes the problem of policy learning as a multi-task learning of context-specific policies. A time-varying and context-dependent policy linearly maps the patient symptoms to treatment. The context updates with new symptoms and treatments. Effectively CPR is learning a linear behavior cloning plicy whose model parameters adapt to history information, with the history affecting the model parameters instead of the input, thus retaining interpretability. Experiments with a simulator show that CPR can converge to the true decision model parameters. On two behavior cloning tasks with real medical data, it matches the performance of black-box models in terms of recovering best treatments.

**Strengths:**

1. proposed a novel approach to interpretable behavior cloning
2. Thorough experimentation and analyses

**Weaknesses:**

1. the interpretability is only retained for the current observation in the logistic regression but not for history values, nor the dynamics (how the values progress across time), both of which can be more important than some of the current values in real-world problems. Indeed the authors did mention that 'CPR assumes that a physician places the highest importance on the most current patient information when deciding an action', but this however contradicts their own problem setting of the partially observable environment in which 'the action probability ... is a function of ... current and past patient symptoms, as well as past actions'. So it seems to me either limited interpretability in POMDPs (without interpreting history), or interpretability assuming the current observation is most important to and suffice to interpret the action (which is no longer an unconstrained POMDP). I'm not so convinced how valuable either is.
2. the authors appear to be using behavior cloning and imitation learning interchangeably.
3. since patients with missing observation values were excluded, it's hard to say how well the model can make use of historical information, i.e., the environment was not partially observable enough.
4. (minor) typo in Eq. 1 $P_w(\theta|c)$

**Questions:**

1. how did you realize the stochasticity around $\theta$ suggested in Eq. 1 if it's parameterized only by an RNN?
2. why all the experimented environments have a binary action space? It'd be good to see how the interpretability and accuracy generalize to multi-class predictions with a linear model.
3. as the weights of the regression model are parametrized by an RNN, there is inductive bias between consecutive timesteps. So I would envisage it's possible that e.g. a feature is found to be important at a certain step not least because it was important at the previous step, but not that it's actually still important (or its current importance being exaggerated). Would you think this should be a problem and why?

---

> ### Author Response · Authors · 2023-11-17
> **Response to Reviewer TJfx (1/2)**
>
> We sincerely thank the reviewer for their constructive comments and questions. We have updated the manuscript to improve organization, results interpretation, and add a new experiment in Section 4.2. We thoroughly address your concerns below.
>
> 1. __Interpretability of CPR. There is no free lunch.__
>
> While global interpretability may be essential in some domains, we collaborate extensively with clinicians to design CPR explicitly for medical imitation learning, a burgeoning field where data has rapidly outpaced methods development and new solutions are desperately needed. From our collaboration with clinicians on this project, we identify that current symptoms are almost always the most important features for treatment, while patient history serves to place these current observations _in context_. As Reviewer 9ccK mentioned, there is no free lunch. We design CPR to provide meaningful results and interpretation on features that directly affect decisions while limiting our interpretability on features that do not require direct interpretation. This is significant to both the medical informatics and machine learning communities. CPR provides a tradeoff between two absolutes (black-box models versus globally interpretable models) through a novel representation framework that enables exact interpretability of key features where previously there was no middle ground. This conceptual framework is generally adaptable and provides machine learning researchers with a new tool to enable direct interpretability where it matters without sacrificing accuracy. As such, CPR is not designed for unconstrained POMDPs, but makes use of a common real-world constraint to identify a general case where actions are mediated through history-dependent models and presents a novel representation to capture and interpret this dependence. This is especially valuable to the fast-growing field of medical imitation learning, but is generally applicable for understanding and interpreting context-dependent effects on agent actions. To bring attention to the benefits and drawbacks of this tradeoff, we have added a caveat in our updated discussion section about when the context-specific policy is a bug and when it is a feature.
>
> Furthermore, black-box models are generally thought of as globally optimal policy learners with ideal representations, although these representations are uninterpretable. We design a simulation study, now included in section 4.2 (results in Figure 5) to explore how black-box representations compare to CPR’s explicit context-specific policy representation. Indeed, we identify a notable limitation of black-box methods in representing this type of POMDP and show CPR’s superior performance on representation learning as well as action-matching accuracy. We encourage the reviewer to read our response to Reviewer KxoJ for a brief overview of this experiment and look at the updated manuscript for the full rigorous evaluation. We believe this result has significance to the overall representation learning community and underscores the need for explicit and interpretable policy representations that align with problem constraints.
>
> 2. __Behavior cloning and imitation learning are used interchangeably__
>
> Thank you for alerting us to this ambiguity. Behavior cloning is a specific implementation of imitation learning based on supervised learning that emphasizes black-box policy performance, while imitation learning methods in general focus on policy reasoning but are not tied to a specific optimization scheme. CPR spans both areas, emphasizing both policy reasoning and performance. Based on prior literature terminology [1] and CPR’s interpretable design, we call CPR a medical imitation learning method and have updated the manuscript to be more consistent.

---

> ### Author Response · Authors · 2023-11-17
> **Response to Reviewer TJfx (2/2)**
>
> 3. __Realization of the stochasticity around $\theta$__
>
> Thanks for the suggestion! We have clarified the problem setup in Section 3.1, and we conducted an additional experiment to discuss the robustness of our model with stochastic $\theta$. We find that CPR’s novel context-specific policy representation is significantly more robust to noise than black-box methods, with the context-specific policy serving as both an interpretability mechanism and a regularizer (Figure 5). We believe this result will be of interest to the representation learning and interpretable ML communities.
>
> 4. __All the experimented environments have a binary action space.__
>
> The multi-action setting is possible with CPR, but we design our study with an emphasis on binary actions to enable fair comparison with recent baseline methods [1, 2], which have significant reproducibility issues.
>
> The most notable methods for medical imitation learning either require expert knowledge for implementation on new tasks [2] or are completely inaccessible [1] with dead links to supposedly open-source code. For these reasons, we have been unable to reproduce these supposedly SOTA baselines for new medical imitation learning tasks, which might extend to a multi-action setting. Furthermore, there are major discrepancies between these works in their reported results (see the INTERPOLE results in Table 2 from both [1] and [2]), and in many cases both are outperformed by a simple logistic regression baseline, and their reported comparison against a vanilla RNN is not rigorous, since we achieve a far stronger RNN baseline with minimal hyperparameter tuning (our Table 2 RNN vs. the BC baselines in [1] and [2]). This inconsistency and lack of reproducibility is a major concern for the fast-growing medical imitation learning community. In response, we have made CPR open-source and designed it to be task-agnostic, allowing an easy and accessible baseline for future work. The lack of reproducible baselines also limited our comparison with prior works to datasets where results were published and preprocessing code was standardized. There is an immediate and pressing need to publish open-source and adaptive tools to foster growth in this community. Alongside CPR’s positive results, our negative results on the reproducibility of prior works and CPR’s open-source and task-agnostic code are a timely and important contribution to this community.
>
> 5. __Inductive bias of RNN and LSTM__
>
> Thank you for raising this point. Our core framework splits black-box and glass-box models into modular components to enable context-specific interpretability. As a secondary consequence, any recurrent model can be substituted for the black-box model. We explore the RNN and LSTM for proof of concept, and our results strongly support that CPR is SOTA in terms of both accuracy and interpretability with this implementation. We primarily focus on validating the utility and performance of our novel interpretability mechanism, the context-specific policy. Extensive exploration of inductive biases in RNNs and LSTMs would distract from this aim, and we leave this to future work given that our results already strongly support CPR’s SOTA performance in terms of both accuracy and interpretability.
>
> __Misc__
> - Math typos have been fixed and clarified.
>
> Overall, we believe that CPR is a timely and important method for the nascent field of medical imitation learning, and we have presented a rigorous evaluation from both machine learning and clinical perspectives. We have thoroughly addressed the reviewer’s comments and added additional experiments to set a new standard for results presentation in this domain. Based on this discussion and new information, as well as CPR’s SOTA performance on success metrics, and it’s data-driven discovery of nuanced treatment strategies that were missed by previous works, we hope the reviewer’s score can be updated to reflect the significance, timeliness, and rigor of our study within the medical imitation learning and broader representation learning communities.
>
> [1] Pace, A., Chan, A., & Schaar, M. van der. (2022, January 28). POETREE: Interpretable Policy Learning with Adaptive Decision Trees. International Conference on Learning Representations. https://openreview.net/forum?id=AJsI-ymaKn_
>
> [2] Hüyük, A., Jarrett, D., Tekin, C., & Schaar, M. van der. (2022, February 10). Explaining by Imitating: Understanding Decisions by Interpretable Policy Learning. International Conference on Learning Representations. https://openreview.net/forum?id=unI5ucw_Jk

---

> ### Comment · Reviewer_TJfx · 2023-11-21
>
> Thank you authors for your response.
>
> 1. Thanks for acknowledging that CPR is not designed for unconstrained POMDPs. My concern is that it could be worse in cases where the current action was in fact partly attributed to history whilst you are forcing it to be fully explained by current observation only which could be extrapolated/erroneous. Even in medicine can you guarantee an action can always be interpreted (not just determined) by the latest observation?
>
> 4.Lack of open source implementation of baselines doesn't seem sufficient to me to justify omitting this critical evaluation, as there should be significant discrepancy between the mechanisms of binary and multiclass classifications. We can alway code up ourselves.
>
> 5.I don't think this question is being answered. Given that everything is modular, the paper only experimented with RNN and LSTM, so from the results we cannot tell whether the temporal inductive bias (in terms of action interpretability not of history encoding btw) was contributing positively/negatively/nothing.
>
> I'll keep my score.

---

> ### Author Response · Authors · 2023-11-22
> **Follow-up Response to Reviewer TJfx**
>
> We thank the reviewer for their comment and address their concerns below.
>
> 1\. On the topic of modeling assumptions, all models are wrong but some are useful. We can trivially construct similar failure scenarios for the baseline models in our study. All modeling assumptions will be violated in some cases. However, good assumptions should align with real-world mechanisms to minimize representation failures. Rather than focusing on hypothetical failure cases, we conduct a data-driven evaluation of our model's validity by assessing the alignment between real-world evidence and an CPR's policy explanations. Indeed, we find that CPR's data-driven explanations align with real medical literature and known treatment practices (Figures 2, 3, 4), providing nuanced and medically meaningful interpretations that were missed by prior works due to sub-par model performance and alignment (Figures 3, 4). We cannot guarantee CPR's assumptions will never be violated, and we note this in our discussion section. However, our experiments strongly support that CPR achieves SOTA interpretability by improving the alignment of its representation with the mechanisms governing real world medical decisions.
>
> 4\. As noted in our summary comment to all reviewers, despite being published at ICLR the main baselines are irreproducible on top of being closed-source (e.g. outperformed by logistic regression in our experiments even though they claimed to outperform LR, Table 2 in ours vs. Table 2 in [1] and [2]). Hence, our approach to benchmarking CPR was to measure performance on exactly the same datasets that were used in the prior works. This approach allowed a fair comparison across the open-source CPR and closed-source baseline methods. In this way, we confirm that CPR is SOTA, achieving a dramatic and significant performance improvement on both real-world datasets (Table 2, now updated to show standard errors). The only discrepancy between our evaluation and prior works is that prior works did not report AUPRC for the MIMIC antibiotic-prescription task. However, this did not prohibit the recent publication and spotlight presentation of these prior works at ICLR. By including this metric in our evaluation of CPR and open-source baselines, we strengthen our study and improve the impact of our results reporting over prior studies.
>
> None of the previous works have discussed the multi-action setting. Requiring CPR to perform benchmarking experiments that go beyond what was reported by these prior ICLR papers is inappropriate. For reviewers aware of the closed-source and irreproducible nature of the baseline models, insisting on such requests seems unethical.
>
> 5\. In extensive experiments on real and simulated data, we confirm that CPR provides SOTA accuracy and interpretability. We test CPR with both RNN and LSTM black-box components, and find that either implementation would be considered SOTA. There is no meaningful scenario where exploring the effect of the temporal inductive bias would change our results. Our experimental evidence is sufficient, and strongly indicates CPR's significance to the medical informatics, representation learning, and XAI communities at ICLR. We hope that the reviewer's score can be updated to reflect this.

---

### Official Review · Reviewer_9ccK · 2023-10-31

**Soundness:** 2 fair
**Presentation:** 2 fair
**Contribution:** 1 poor
**Rating:** 3
**Confidence:** 3

**Summary:**

This paper is concerned with learning interpretable policies from data. Different from prior work, it frames the problem as learning context-specific interpretable policies. The paper claims that instead of a global interpretable policy, having multiple contextualized interpretable policies results in an improved tradeoff between accuracy and interpretability. Each context-specific policy is called interpretable because it induces a linear mapping from observations to actions. On the other hand, these interpretable policies are parameterized by contexts, which can be generated from historical data using black-box models. The paper’s claims were validated by two experiments on medical domains. One of them concerns interpreting MRI prescriptions. The other one is about predicting antibiotic prescriptions.

**Strengths:**

With the growing use of machine learning methods in high-stakes real-world applications, questions posing the interpretability of machine learning methods have become a serious concern in the field. This paper tries to improve the tradeoff between interpretability and accuracy by proposing contextualized policy recovery. AUROC improvements in the accuracy compared to SOTA interpretable policy learning methods are promising. It is also good that the authors focus their experimental results and discussion on two real-world medical problems that intrigue researchers and practitioners.

**Weaknesses:**

I am not convinced by the originality and significance of the proposed approach. On the one hand, the proposed approach comes with remarkable performance, almost matching those of non-interpretable black-box counterparts. On the other hand, it deems itself as being as interpretable as previous interpretable policies. Reasoning dictates that there should be no free lunch. Looking deeper into the proposed contextualized policy recovery (CPR), I see it as a grey box rather than a glass box. It is a glass box only for the current context, which is good. However, all the past information, including previous contexts and actions, is hidden inside a non-interpretable black-box context encoder. I think that this jeopardizes the interpretability of the envisioned medical applications. To put things into context, given the same context at time t, different action probabilities can be recommended based on the history-dependent parameterization. Ideally, we want to understand which contexts and actions in the past caused this difference. However, if the context encoder is black-box, it sheds no light on why this difference happened. I think that this is a big obstacle to interpretability.

**Questions:**

The experiments mainly involve applying CPR to medical datasets and interpreting the resulting context-specific decision models. In line with the main premise of CPR, decision functions change under different contexts. I am confused about the interpretation of the heterogeneity of the decision policies. For instance, for the MIMIC antibiotics dataset, it is concluded that the main driver of heterogeneity is “prior antibiotic prescription”. Is it a conclusion that CPR reached? This should be the case if CPR produces interpretable policies. However, it does not seem to be the case since prior antibiotic prescription (which is an action taken in the past) is mapped to a context by a black-box context encoder.

---

> ### Author Response · Authors · 2023-11-17
> **Response to Reviewer 9ccK (1/2)**
>
> We sincerely thank the reviewer for their constructive comments. We have updated the manuscript to improve organization, results interpretation, and add a new experiment in Section 4.2. We thoroughly address your concerns below.
>
> 1. __The novelty and significance of CPR__
>
> CPR achieves SOTA accuracy (Table 2) and interpretability (Figures 2, 3, 4, 5), significantly improving on previous medical imitation learning methods. These results strongly support the originality and significance of our work.
>
> Quantitative results indicate SOTA empirical performance, +7% and +22% AUROC on canonical tasks with standardized success metrics versus other interpretable models. We then validate CPR’s performance by assessing alignment of context-specific policies (our novel interpretability mechanism and policy representation) against medical literature in collaboration with clinical collaborators. In this follow-up, we present a variety of data-driven evaluations, all of which directly utilize CPR’s mechanism of interpretability and reproduce best-practice treatment regimes without prior knowledge. Indeed, CPR recovers meaningful heterogeneity (age and gender risk-factors, Fig. 4), context dependence (prior treatments, Fig. 2), and outlier scenarios (kidney failure, Fig. 3) driving medical decisions in context-specific policies from data alone, which often were missed by prior works on medical imitation learning [1, 2] due to sub-par model alignment and performance.
>
> We further improve our study’s relevance to the general machine learning community with a new simulation study and empirical evaluation, detailed in Section 4.2 and Figure 5 in the revised manuscript. We explore how black-box policy representations compare to CPR’s explicit context-specific policy representation. In this experiment, we identify notable limitations of black-box representations and subsequent performance on this type of context-dependent Markov decision process. We show CPR’s superior performance on representation learning as well as action-matching accuracy under a variety of stringent simulation cases. We encourage the reviewer to read our response to Reviewer KxoJ for a brief overview of this experiment and look at the updated manuscript for the full rigorous evaluation. We believe this result has significance to the overall representation learning community and underscores the significance of CPR’s novel context-specific policy representation.
>
> 2. __The interpretability of CPR. There is no free lunch.__
>
> We developed CPR in collaboration with physicians to make accurate and realistic assumptions about clinical decision making. Through this collaboration, we identified the importance of context in interpreting medical observations, and the lack of existing methods that encode context-dependence in decision policies. Consequently, no existing methods are both accurate and interpretable for medical imitation learning. In CPR, we address this by designing a machine learning architecture with a novel interpretable representation of medical decisions: the context-specific policy. We directly utilize the learned context-specific policy representations to provide nuanced, data-driven interpretations of medical decisions, which are extensively validated by medical literature (Sections 4.1.1 and 4.1.2, Figures 2, 3, 4). In summary, CPR’s interpretability mechanism is strongly supported by experimental evidence and clinical intuition.
>
> Furthermore, CPR’s design and explicit tradeoff between global and local interpretability has led to a novel mechanism of adaptive interpretability that has significance to the broader machine learning community, which we detail in our first discussion point with Reviewer TJfx.

---

> ### Author Response · Authors · 2023-11-17
> **Response to Reviewer 9ccK (2/2)**
>
> 3. __Discovery of main driver of MIMIC treatment decisions as prior antibiotic prescription__
>
> CPR includes all historical information in its context encoding, including actions taken at prior time steps i.e. antibiotic prescription. In this way, CPR continuously revises its context-specific policies with the most up-to-date information about past states and actions, inferring dynamic and patient-specific policy models. This adaptability is a key feature, and enables us to recover highly heterogeneous policies across our patient populations. We do not provide CPR with any prior domain-specific knowledge about medical decisions. After training, we use CPR to infer a series of context-specific policy coefficients for each patient. We then analyze these policy coefficients to identify features which differentiate policy coefficient trajectories. For instance, Figure 2 shows the organization of antibiotic prescription policies over time into two primary clusters, which are differentiated by prior antibiotics prescription. Appendix Figure 10 shows the variation of antibiotic prescription policies with respect to other patient symptoms, and Appendix Figure 11 shows a clustering of all patient trajectories which identify several common treatment regimes. All of CPR’s interpretations (Figures 2, 3, 4) are based on this data-driven analysis of policy coefficients, which reveal medically-validated  best-practice treatment plans _without any prior knowledge_, including the importance of prior antibiotics prescription in future antibiotics prescription [3] and risks of antibiotic use [4].
>
> Overall, we believe that CPR is a timely and important method for the nascent field of medical imitation learning, and we have presented a rigorous evaluation from both machine learning and clinical perspectives. We have thoroughly addressed the reviewer’s comments and added additional experiments to set a new standard for results presentation in this domain. Based on this discussion and new information, as well as CPR’s SOTA performance on success metrics, and it’s data-driven discovery of nuanced treatment strategies that were missed by previous works, we hope the reviewer’s score can be updated to reflect the significance, timeliness, and rigor of our study within the medical imitation learning and broader representation learning communities.
>
>
> [1] Pace, A., Chan, A., & Schaar, M. van der. (2022, January 28). POETREE: Interpretable Policy Learning with Adaptive Decision Trees. International Conference on Learning Representations. https://openreview.net/forum?id=AJsI-ymaKn_
>
> [2] Hüyük, A., Jarrett, D., Tekin, C., & Schaar, M. van der. (2022, February 10). Explaining by Imitating: Understanding Decisions by Interpretable Policy Learning. International Conference on Learning Representations. https://openreview.net/forum?id=unI5ucw_Jk
>
> [3] Guleria, R., Mohan, A., Kulkarni, A., Madan, K., & Hadda, V. (2019). Guidelines for Antibiotic Prescription in Intensive Care Unit. Indian Journal of Critical Care Medicine, 23(S1), 1–63. https://doi.org/10.5005/jp-journals-10071-23101
>
> [4] Khalili, H., Bairami, S., & Kargar, M. (2013). Antibiotics induced acute kidney injury: Incidence, risk factors, onset time and outcome. Acta Medica Iranica, 51(12), 871–878.

---

### Official Review · Reviewer_KxoJ · 2023-11-06

**Soundness:** 2 fair
**Presentation:** 3 good
**Contribution:** 2 fair
**Rating:** 5
**Confidence:** 4

**Summary:**

The paper proposes "contextual" policy recovery, a new interpretable method for parametrizing policies for imitation learning to mitigate the interpretability x accuracy tradeoff. The parametrization uses interpretable policy functions for $\pi_\theta(a_t \mid x_t)$ such as a linear class and summarizes historical patient context through the parametrization of this policy class i.e., $\theta = g(x_{1:t-1}, a_{1:t-1})$. The authors demonstrate that this parametrization improves over current imitation learning baselines in terms of the accuracy/interpretability tradeoff. Empirical evaluation in MIMIC (and ICU healthcare dataset for antibiotics treatment) and ADNI MRI scan data for cognitive disorder diagnosis suggests improved tradeoffs. Authors have tried to exhaustively interpret the results from both datasets.

**Strengths:**

1. Interpretability of learned offline RL policies is crucial for safety of implementation in applications such as healthcare. Authors are trying to improve the accuracy interpretability tradeoff by proposing a new parametrization of the policy class.

2. The paper is well motivated and the presentation makes the contributions clear

3. Detailed empirical evaluation of healthcare data, which seems like the main application area of interest, is presented.

4. I think Figure 7 is the most informative in terms of understanding how contextualized policy recovery helps without needing significant domain expertise.

**Weaknesses:**

1. While the contribution is simple and easy to understand, I believe the contribution is a bit incremental than one would like.

2. I think its a strong assumption to make that imitation learning is the best thing to do in MIMIC and ADNI datasets. Healthcare biases in EHR data are fairly well-reported [1], although one might hypothesize it is less so for ICU data. Without clear discussion of why this is a reasonable choice, the relevance and applicability diminishes a bit, in my view.

3. It is quite challenging for me to validate each of the interpretations of the emprical evaluation presented in the paper (beyond simple interpretation like expected values of the learned co-efficients at a very coarse level). A much better empirical evaluation would have been to ask clinicians specific questions about the learned policies and reported the summary statistics on those. I understand that this is challenging to do in generality, but even annotations from one to two clinicians would suffice. At this point because of the depth of emphasis on interpreting the policies, and my lack of clinical background, I am not able to asses most of the empirical evaluation.

4. I am not sure the simulations in 4.2 really add much insight.

[1] Verheij, Robert A., Vasa Curcin, Brendan C. Delaney, and Mark M. McGilchrist. "Possible sources of bias in primary care electronic health record data use and reuse." Journal of medical Internet research 20, no. 5 (2018): e185.

**Questions:**

1. Do the authors have a clinician on the team who assisted with these interpretations?

2. Why haven't authors re-created and reproduced their own results for MIMIC Antibiotics for AUPRC and Brier Score?

3. Why use MIMIC-II when we're already in a world with updated version of MIMIC-IV with a larger number of patients.

4. I don't clearly understand the ADNI MRI task. What is the goal and what is the desired outcome? Can the authors add more context in whether these cognitive impairment rating decisions tend to be accurate in the data?

5. I would suggest adding additional empirical evaluation in more traditional imitation learning tasks to show utility without requiring significant domain knowledge to interpret all results.

6. I would also consider improving the empirical evaluation altogether. Suppose clinicians given you some domain constraints that need to be satisfied (e.g., antibiotics dose should not exceed a certain amount, although this is not the exact decision point here). Can the authors demonstrate that using contextualized policy learning more easily satisfies such constraints (without explicitly adding them) compared to existing baselines and RNN/LSTM parametrizations of the policy class? I think this would add some interesting evaluation than is currently presented in the paper.

---- Updates based on author responses--------------------------------
Thank you for the comments and for clarifying my concerns. It sounds like the authors claim is that this kind of imitation learning is useful to uncover biases in healthcare data. If that is the case, my recommendation is that the motivation needs to be updated to reflect what the contribution is.

I still don't know why use MIMIC-III over MIMIC-IV

I also don't believe methods like LIME can actually provide real interpretability of treatment policies in a way that's actionable to clinicians.

I do believe adding more clinician annotations will make the study interesting and the empirical evaluation more credible.

Based on the author responses, I have improved the score.

---

> ### Author Response · Authors · 2023-11-16
> **Response to Reviewer KxoJ (1/4)**
>
> We thank the reviewer for their time and constructive comments. We have updated the manuscript to improve organization, results interpretation, and add a new experiment in Section 4.2. We thoroughly address your concerns below.
>
> 1. __Incremental contribution__
>
> We provide an objective and domain-agnostic empirical evaluation of CPR against prior works on standardized success metrics in Table 2, and these quantitative results indicate extremely strong performance with +7% and +22% AUROC on canonical tasks for medical imitation learning. We then qualify and validate CPR’s performance on these metrics by comparing CPR’s interpretable context-specific policies against medical literature on treatment policies. In this follow-up, we present a variety of data-driven evaluations that directly utilize CPR’s mechanism of interpretability (Figures 2, 3, 4) to discover nuanced treatment policies and these treatment policies that are strongly supported by medical literature. Through rigorous quantitative and qualitative evaluation, we show that the context-specific policies generated by CPR are both medically meaningful and highly predictive. We directly utilize the learned context-specific policy representations to provide nuanced data-driven interpretations of medical decisions, which are extensively validated by medical literature (Figures 2, 3, 4). CPR recovers treatment heterogeneity relating to age and gender-based risk-factors (Fig. 4) [3], the context dependence of prior treatments (Fig. 2) [4], and catastrophic outlier scenarios such as kidney failure (Fig. 3) [5]. Our interpretable framework reveals driving factors of medical decisions that were missed by prior medical imitation learning methods [1, 2] and would be impossible to discover with these methods due to sub-par model alignment, lack of context dependence, and low accuracy.
>
> We further improve our study’s relevance to the general machine learning community with a new simulation study and empirical evaluation, detailed below and available in the re-uploaded manuscript. To summarize, our work has significance to both the representation learning, interpretable ML, and medical informatics communities, and our results strongly support that CPR is SOTA in terms of both accuracy and interpretability for medical imitation learning.
>
> 2. __Clinical collaborations and expert-in-the-loop design__
>
> We developed CPR in collaboration with physicians to make accurate and realistic assumptions about clinical decision making. Through this collaboration, we identified the importance of context in interpreting medical observations, and the lack of existing methods that encode context-dependence in decision policies. In CPR, we address this by designing a machine learning architecture with a novel interpretable representation of medical decisions: the context-specific policy, which we rigorously evaluate using real and simulated data as well as medical domain knowledge.
>
> 3. __Decision to use MIMIC-III and ADNI datasets for imitation learning__
>
> Electronic record keeping is ubiquitous in healthcare, but data collection has vastly outpaced the development of methods to analyze, quantify, and provide actionable data-driven insights about clinical operations. To address this, medical imitation learning aims to quantify and explain the decisions physicians make during treatment, providing an unprecedented opportunity to identify biases in treatment, unexplained variability among patients, and improve clinical operations. For precisely the reasons the reviewer mentions (EHR and treatment biases, domain knowledge necessity) as well as heterogeneity among patients, medical applications are perhaps the most difficult yet important use case of imitation learning. Methods designed specifically for _medical_ imitation learning are needed to address the unique nuances and constraints of medical data [1, 2]. Although the reviewer rightfully points out systemic biases in EHR data collection, the MIMIC and ADNI datasets used in our evaluation have become canonical tasks for medical imitation learning methods based on their well-understood clinical workflows, known biases and nuances to treatment, and the general consensus among clinicians that real policies can be improved for these tasks. When biases in treatment are present, CPR can recover these biases. CPR effectively detects age and gender-driven biases in treatment (Figure 4) which are validated by medical literature [3, 4, 5].

---

> ### Author Response · Authors · 2023-11-16
> **Response to Reviewer Kxoj (2/4)**
>
> 4. __Baseline reproducibility__
>
> The most notable methods for medical imitation learning either require expert knowledge for implementation on new tasks [2] or are completely inaccessible [1] with dead links to supposedly open-source code. For these reasons, we have been unable to reproduce these supposedly SOTA baselines for new medical imitation learning tasks. Furthermore, there are major discrepancies between these works in their reported results (see the INTERPOLE results in Table 2 from both [1] and [2]), and in many cases both are outperformed by a simple logistic regression baseline, and their reported comparison against a vanilla RNN is not rigorous, since we achieve a far stronger RNN baseline with minimal hyperparameter tuning (our Table 2 RNN vs. the BC baselines in [1] and [2]). This inconsistency and lack of reproducibility is a major concern for the fast-growing medical imitation learning community. In response, we have made CPR open-source and designed it to be task-agnostic, allowing an easy and accessible baseline for future work. The lack of reproducible baselines also limited our comparison with prior works to datasets where results were published and preprocessing code was standardized. There is an immediate and pressing need to publish open-source and adaptive tools to foster growth in this community. Alongside CPR’s positive results, our negative results on the reproducibility of prior works and CPR’s open-source and task-agnostic code are a timely and important contribution to this community.
>
> 5. __Domain-agnostic evaluation of CPR__
>
> Given that both primary baselines were published recently at ICLR, and that both baselines only contain studies with real observational data and a similar simulation to our previous result, we believe that it would be inappropriate and unfair to reject this work on the basis that it does not include a more domain-agnostic evaluation. However, we do agree with the reviewer that performing objective assessment of medical imitation learning methods on a known MDP would push the field forward and set an excellent example for future work. To foster the development of interpretable imitation learning methods and promote comparisons with black-box models, we design a simulation study to compare CPR against black-box models and post-hoc interpretability methods. We evaluate CPR and black-box methods in terms of action-matching accuracy and their recovery of known policy parameters. In this simulation, we find that CPR’s SOTA performance and interpretability is driven by its explicit representation of the context-specific policy, which in certain cases improves over the performance of even black-box models, a significant result for the representation learning and interpretable ML communities. This simulation study is in the new Section 4.2 with results in Figure 5, but for the reviewer’s convenience we summarize below.

---

> ### Author Response · Authors · 2023-11-16
> **Response to Reviewer KxoJ (3/4)**
>
> __Improved Simulation Study: Contextualized Policies vs. Post-hoc Interpretability__
>
>
> We aim to test CPR in known environments (simulations) to understand how its learned policy representation deviates from true policies. Toward this end, we provide an improved simulation and a comparison with post-hoc interpretability methods. CPR incorporates both a deep learning component and a statistical modeling component to introduce a novel mechanism of interpretability, the context-specific linear policy. Our approach differs substantially from prior interpretable methods by including a deep learning component. Naturally, we explore if CPR's explicit linear policy representation is key to its performance and interpretability, or if accurate and robust context-specific linear policies can also be recovered from black-box policy models using post-hoc interpretation methods. To test this, we simulate a heterogeneous, action-dependent Markov decision process (MDP) and evaluate CPR versus black-box baselines on their recovery of true simulation parameters: the true action probability and the true coefficients of a context-specific linear policy. While CPR explicitly generates these context-specific linear coefficients, black-box models implicitly model these coefficients as feature gradients (i.e. linear coefficients in a first-order Taylor expansion). Akin to popular post-hoc interpretability methods like LIME by (Ribeiro et al., 2016), we leverage the differentiability of RNNs $\Phi(x_t, h) \rightarrow a_t$ to recover the implicit context-specific linear policies $\theta$. $$\widehat{\theta} = \frac{\partial}{\partial x_t}\Phi(x_t, h)$$ We generate data with a known heterogeneous MDP governed by a true context-specific policy $$P(a_t = 1|x_t, x_{t-\tau}, a_{t-\tau}, t) = 1 / (1 + \exp(-\theta \cdot x_t + \epsilon_a))$$ $$\theta = x_{t-\tau} \cdot (2 a_{t-\tau} - 1) + \frac{t}{T} + \epsilon_{\theta}$$ where $T$ is the sequence length or maximum timestep, and $\tau$ is a time lag between the current policy and a dependence on past observations $x_{t-\tau}$ and actions $a_{t-\tau}$. We simulate $N$ total sequences, drawing observations $x_t \sim \operatorname{Unif}[-2, 2]$, decision noise $\epsilon_a \sim N(0, \sigma_{a}^2)$, and policy noise $\epsilon_{\theta} \sim N(0, \sigma_{\theta}^2)$ at each timestep. We compare the policy models learned by CPR and RNN in terms of the Pearson's correlation coefficient (PCC) between estimated and true action probabilities and context-specific policy coefficients. We hold out 15% of trajectories at random for evaluation. Results for total patients $N=200$, time-lag $\tau = 2$, decision noise $\sigma_a = 0$, and $\sigma_{\theta} = 0$, and sequence length $T = 10$ are below, presented below as mean (std).
>
> |              | PCC θ            | PCC P(a)        |
> |--------------------|------------------|-----------------|
> | CPR explicit policy model | 0.830 (0.036)    | 0.848 (0.029)   |
> | RNN implicit policy model | 0.203 (0.021)    | 0.706 (0.087)   |
>
> On this known heterogeneous and action-dependent MDP, CPR's explicit policy representation not only improves its representation of MDP parameters but increases overall performance versus a black-box model with an unstructured policy representation. It is exciting to us that CPR's policy representation is not only more accurate than implicit policy representations in unstructured black-box models, but that CPR's representation choice also serves as a regularizer under the right circumstances to improve prediction accuracy as well.

---

> ### Author Response · Authors · 2023-11-16
> **Response to Reviewer KxoJ (4/4)**
>
> We believe that CPR is a timely and important method for the nascent field of medical imitation learning, and we have presented a rigorous evaluation from both machine learning and clinical perspectives. We have thoroughly addressed the reviewer’s comments and added additional experiments to set a new standard for results presentation in this domain. Based on this discussion and new information, as well as CPR’s SOTA performance on success metrics, and it’s data-driven discovery of nuanced treatment strategies that were missed by previous works, we hope the reviewer’s score can be updated to reflect the significance, timeliness, and rigor of our study within the medical imitation learning and broader representation learning communities.
>
> Misc:
> - MIMIC-II was a typo in the original version, we study CPR on MIMIC-III.
> - The ADNI MRI task is predicting when a physician will prescribe an MRI for diagnosing Alzheimer’s. In medical imitation learning, we do not directly model the patient outcome. Instead, we aim to understand when a physician makes a decision and why. MRIs are expensive, time-consuming, and uncomfortable, and over-prescription and slow hospital operations and place patients under unnecessary stress. Data-driven evaluation of MRI prescription policies seeks to improve hospital operations and patient experiences.
> - Domain constraints would be excellent in practice, but in this study we design CPR to be domain-agnostic within medical imitation learning. We show that this approach recovers nuanced and medically valid treatment regimes with minimal prior bias, allowing us to validate CPR’s interpretations against known best-practices. We believe this minimal-bias approach to data-driven policy modeling is the most adaptable and convincing framework for our study.
>
>
> __References__
>
> [1] Pace, A., Chan, A., & Schaar, M. van der. (2022, January 28). POETREE: Interpretable Policy Learning with Adaptive Decision Trees. International Conference on Learning Representations. https://openreview.net/forum?id=AJsI-ymaKn_
>
> [2] Hüyük, A., Jarrett, D., Tekin, C., & Schaar, M. van der. (2022, February 10). Explaining by Imitating: Understanding Decisions by Interpretable Policy Learning. International Conference on Learning Representations. https://openreview.net/forum?id=unI5ucw_Jk
>
> [3] Castro-Aldrete, L., Moser, M. V., Putignano, G., Ferretti, M. T., Schumacher Dimech, A., & Santuccione Chadha, A. (2023). Sex and gender considerations in Alzheimer’s disease: The Women’s Brain Project contribution. Frontiers in Aging Neuroscience, 15. https://www.frontiersin.org/articles/10.3389/fnagi.2023.1105620
>
> [4] Guleria, R., Mohan, A., Kulkarni, A., Madan, K., & Hadda, V. (2019). Guidelines for Antibiotic Prescription in Intensive Care Unit. Indian Journal of Critical Care Medicine, 23(S1), 1–63. https://doi.org/10.5005/jp-journals-10071-23101
>
> [5] Khalili, H., Bairami, S., & Kargar, M. (2013). Antibiotics induced acute kidney injury: Incidence, risk factors, onset time and outcome. Acta Medica Iranica, 51(12), 871–878.

---

### Author Response · Authors · 2023-11-21
**Summary Response to Reviewers and AC**

We provide a summary response to reviewers and the AC here.

CPR (our proposed method) is a new method for interpretable policy recovery. Interpretable policy recovery is an important task in medical informatics and a few datasets have emerged as the community standard. We test CPR against two of these standard datasets and achieve new SOTA performances: +22% AUROC on the antibiotic prescription task and +7% AUROC on the MRI prescription task.

Not only does CPR exhibit SOTA accuracy in these policy recovery tasks, it also improves interpretability of the learned policies. In these standard datasets, CPR reveals new insights that have previously been missed by all prior publications. We believe this combination of SOTA accuracy and interpretability make CPR worth sharing to the community.

__Fit to ICLR community:__ The two most relevant baseline models [1,2] were published at ICLR and honored with spotlight presentations. CPR significantly outperforms both models. Furthermore, in developing CPR, we discovered critical limitations of these baseline models (including that one of the models is actually outperformed by logistic regression). Hence, we believe the ICLR community would be interested in these results.

__The novelty of CPR compared to baselines:__ CPR proposes a fundamentally new view of medical decisions: that policies are inherently simple when understood in proper context. Hence, CPR proposes to learn a function mapping context features to context-specific policies. This is a new view of policy learning that departs from prior models built on assumptions of universal [1, 2, 3, 4, 5] policies. As a result of this new view and new architecture, CPR significantly outperforms baseline models on standard benchmarking datasets. (+22% AUROC on the antibiotic prescription task and +7% AUROC on the MRI prescription task). We believe this new philosophy, new architecture, and SOTA performance attest to the novelty of CPR.

__The interpretability of CPR:__ CPR’s novel architecture allows a new mechanism of interpretability: the context-specific policy. This new mechanism of interpretability reveals new insights about datasets. Even in the standard benchmarking datasets, we find interesting new interpretations including: best-practice treatment guidelines (Figure 2), variability among patients within best-practice treatment guidelines (Figure 4), and scenarios where clinicians avoid treatment to mitigate catastrophic risks (Figure 3). These insights were missed by the baseline models [1, 2], so we strongly believe CPR is at least as interpretable as the baseline models.

__CPR as a community resource:__ Despite being published at ICLR, the main baselines are closed-source and irreproducible (e.g. outperformed by logistic regression in our experiments even though they claimed to outperform LR, Table 2). Hence, our approach to benchmarking CPR was to measure performance on exactly the same datasets that were used in the prior works. This approach allowed a fair comparison across the open-source CPR and closed-source baseline methods.

We cannot expand and stress-test the closed-source, irreproducible baseline models. Hence, requiring CPR to perform benchmarking experiments that go beyond what was reported by these prior ICLR papers is inappropriate. For reviewers aware of the closed-source nature of the baseline models, insisting on such requests seems unethical.

CPR achieves SOTA performance on standard success metrics while also enabling data-driven discovery of nuanced treatment strategies that were missed by previous works. We hope the reviewer’s scores can be updated to reflect the significance, timeliness, and rigor of our study within the medical imitation learning and broader representation learning communities.

__References__

[1] Pace, A., Chan, A., & Schaar, M. van der. (2022, January 28). POETREE: Interpretable Policy Learning with Adaptive Decision Trees. International Conference on Learning Representations. https://openreview.net/forum?id=AJsI-ymaKn_

[2] Hüyük, A., Jarrett, D., Tekin, C., & Schaar, M. van der. (2022, February 10). Explaining by Imitating: Understanding Decisions by Interpretable Policy Learning. International Conference on Learning Representations. https://openreview.net/forum?id=unI5ucw_Jk

[3] Englert, P., Paraschos, A., Deisenroth, M. P., & Peters, J. (2013). Probabilistic model-based imitation learning. Adaptive Behavior, 21(5), 388–403. https://doi.org/10.1177/1059712313491614

[4] Makino, T., & Takeuchi, J. (n.d.). Apprenticeship Learning for Model Parameters of  Partially Observable Environments.

[5] Sun, W., Venkatraman, A., Gordon, G. J., Boots, B., & Bagnell, J. A. (2017). Deeply AggreVaTeD: Differentiable Imitation Learning for Sequential Prediction. Proceedings of the 34th International Conference on Machine Learning, 3309–3318. https://proceedings.mlr.press/v70/sun17d.html

---

### Meta-Review · Area_Chair_ch9d · 2023-12-06

**Metareview:**

This paper seeks to construct interpretable estimates of medical decision policies using a context-dependent perspective. The reviewers appreciated the main motivations of the paper: the need for interpretable policies that do not sacrifice performance compared to black box models. The main concern shared among the reviewers was whether the resulting policy estimates would be interpretable if historical information influences decisions. Though the authors discuss this limitation in the paper (Section 6) and in their discussion with reviewers, stronger evidence/argument is needed to overcome the reviewers' concerns.

**Justification For Why Not Higher Score:**

The reviewers are not convinced that the approach can provide useful interpretations given the limitations in incorporating historical information.

**Justification For Why Not Lower Score:**

N/A

---

### Decision · Program_Chairs · 2024-01-16

Reject